# Reactive Oxygen Species as a Common Pathological Link Between Alcohol Use Disorder and Alzheimer’s Disease with Therapeutic Implications

**DOI:** 10.3390/ijms26073272

**Published:** 2025-04-01

**Authors:** Hyein Song, Jiyong Lee, Yeeun Lee, Seungju Kim, Shinwoo Kang

**Affiliations:** Department of Clinical Pharmacology, College of Medicine, Soonchunhyang University, 31, Soonchunhyang 6-gil, Dongnam-gu, Cheonan-si 31151, Chungcheongnam-do, Republic of Korea; shyein303@gmail.com (H.S.); purple3700@gmail.com (J.L.); yeeunwithdubai@gmail.com (Y.L.); seongjukim9701@gmail.com (S.K.)

**Keywords:** reactive oxygen species (ROS), alcohol use disorder (AUD), Alzheimer’s disease (AD)

## Abstract

Chronic alcohol consumption leads to excessive production of reactive oxygen species (ROS), driving oxidative stress that contributes to both alcohol use disorder (AUD) and Alzheimer’s disease (AD). This review explores how ROS-mediated mitochondrial dysfunction and neuroinflammation serve as shared pathological mechanisms linking these conditions. We highlight the role of alcohol-induced oxidative damage in exacerbating neurodegeneration and compare ROS-related pathways in AUD and AD. Finally, we discuss emerging therapeutic strategies, including mitochondrial antioxidants and inflammasome inhibitors, that target oxidative stress to mitigate neurodegeneration. Understanding these overlapping mechanisms may provide new insights for preventing and treating ROS-driven neurodegenerative disorders.

## 1. Introduction 

Alcohol use disorder (AUD) and Alzheimer’s disease (AD) are two major neurological disorders that share oxidative stress as a key pathological mechanism [1,2]. Chronic alcohol consumption induces excessive production of reactive oxygen species (ROS), leading to mitochondrial dysfunction, neuroinflammation, and neuronal apoptosis—processes that also contribute to AD pathology [3,4]. Despite accumulating evidence linking oxidative stress to both conditions, the precise mechanisms through which alcohol-induced ROS accelerates neurodegeneration remain poorly understood. Recent studies suggest that ROS-driven mitochondrial dysfunction not only exacerbates AUD-related neuronal damage but may also accelerate AD progression by promoting amyloid-beta (Aβ) aggregation and tau hyperphosphorylation [5,6]. Moreover, oxidative stress-induced neuroinflammation appears to be a shared feature of both disorders, implicating immune dysregulation as a potential link between chronic alcohol exposure and neurodegenerative decline [7,8]. Activation of the Toll-like receptor 4 (TLR4) pathway by alcohol metabolites and gut-derived endotoxins, such as lipopolysaccharide (LPS), has been shown to trigger inflammatory cascades that contribute to both AUD-related neurotoxicity and AD pathology (Figure 1) [9,10]. Epidemiological data further support the link between AUD and an increased risk of dementia. Studies indicate that individuals with chronic alcohol abuse have a significantly higher incidence of cognitive impairment and AD-related pathology compared to non-drinkers [11,12]. However, some reports suggest that low-to-moderate alcohol consumption may have neuroprotective effects, underscoring the complexity of alcohol’s impact on brain health [13]. These conflicting findings highlight the need for a deeper investigation into the molecular pathways by which alcohol-induced ROS influences neurodegeneration. This review explores the dual role of ROS in AUD and AD, highlighting mitochondrial dysfunction and neuroinflammation as key shared pathways. We first examine the impact of alcohol-induced ROS on neuronal integrity and immune activation, followed by a comparative analysis of oxidative stress-mediated mechanisms in AUD and AD. Finally, we discuss emerging therapeutic strategies, including mitochondria-targeted antioxidants and inflammasome inhibitors, that hold promise for reducing oxidative damage and slowing neurodegeneration. By bridging insights from both disorders, this review aims to provide a comprehensive perspective on ROS-driven brain pathology and potential therapeutic avenues.

## 2. Reactive Oxygen Species (ROS)

ROS are chemically reactive molecules containing oxygen, primarily including superoxide anions (O_2_^−^), hydrogen peroxide (H_2_O_2_), and hydroxyl radical (•OH). These species form when molecular oxygen interacts with free radicals [19,20,21]. At low levels, ROS function as essential signaling molecules that support cell survival, facilitate cell cycle progression, and regulate processes such as proliferation, differentiation, migration, and programmed cell death. They modulate key pathways—including PI3K/Akt, MAPK, Nrf2/Keap1, NF-κB, and p53—to balance immune responses and maintain redox homeostasis [22,23,24,25,26]. However, excessive ROS can oxidatively damage cellular components—proteins, lipids, and nucleic acids—thereby inducing inflammation and cell death [26]. ROS are generated both endogenously and exogenously. Endogenously, the mitochondrial electron transport chain (ETC) is the principal source, where electron leakage during oxidative phosphorylation (OXPHOS) results in ROS production. Mitochondria contribute approximately 90% of cellular ROS, and mitochondrial dysfunction can amplify ROS levels, promoting oxidative stress and neurodegeneration—a process implicated in AD [4,26,27,28,29,30,31,32]. In addition to mitochondrial electron leakage, exogenous sources such as chronic alcohol consumption also contribute to ROS accumulation. Ethanol metabolism primarily occurs in the liver via alcohol dehydrogenase (ADH), catalase (CAT), and cytochrome P450 2E1 (CYP2E1). Notably, CYP2E1 is highly inducible by alcohol and generates significant amounts of ROS during ethanol oxidation, especially in chronic drinking conditions [33,34,35]. Within the mitochondrial ETC, Complexes I and III are the major sites of ROS production. Complex I, responsible for NADH oxidation and electron entry, is critical for ATP production; its dysfunction is linked to neurodegeneration, amyloid aggregation, and brain atrophy in AD [36,37]. Complex III, due to its high expression, generates ROS that can affect intracellular signaling over longer distances. Deficiency of its subunit RISP has been associated with increased oxidative stress, Aβ accumulation, and other neurodegenerative features, including tauopathy and neuroinflammation [38,39,40]. Exogenously, ROS arise from environmental stressors, such as alcohol, drugs, industrial solvents, certain nutrients, and radiation [41]. For instance, alcohol consumption produces superoxide anions and hydrogen peroxide (H_2_O_2_) in the gastrointestinal tract, contributing to mucosal damage and the development of alcoholic liver disease and pancreatitis [42,43,44]. Moreover, sleep deprivation has been shown to exacerbate oxidative stress—particularly in the hippocampus—by increasing ROS production. This can lead to enhanced Aβ accumulation, abnormal tau phosphorylation, and neuroinflammation, further linking disrupted sleep patterns to AD pathology (Figure 1) [45,46].

## 3. Alcohol, ROS, and Mitochondria

The brain is the most metabolically active organ, consuming large amounts of oxygen primarily for energy metabolism [47,48]. Neurons and astrocytes, which are rich in mitochondria and have high energy demands, are particularly vulnerable to increased production of ROS. Alcohol-induced ROS are generated through several mechanisms. For example, disruption of mitochondrial function during OXPHOS results in excess ROS production [3,4,27]. In addition, although several ethanol-metabolizing enzymes including ADH are expressed in the central nervous system, current evidence suggests that CYP2E1 plays a more prominent role in brain alcohol metabolism, particularly under chronic alcohol exposure, due to its higher inducibility and ROS-generating capacity [49]. With elevated alcohol concentrations or chronic consumption, a greater fraction of alcohol is processed via the microsomal ethanol oxidizing system (MEOS), where CYP2E1 plays a major role in converting alcohol to acetaldehyde while generating ROS—such as superoxide anions and hydrogen peroxide [10,50]. Furthermore, acetylcholine-mediated activation of NADPH oxidases and xanthine oxidase can also contribute to ROS production [51]. Under normal physiological conditions, cells counteract ROS through a complex antioxidant defense system that includes enzymatic antioxidants, such as superoxide dismutase (SOD), glutathione peroxidase (GPx), and catalase (CAT), as well as non-enzymatic antioxidants like glutathione (GSH), which directly neutralizes reactive species and helps maintain redox homeostasis [26]. However, the brain’s defenses are relatively limited due to its high oxygen consumption, the presence of redox-active metals (e.g., iron and copper), its susceptibility to lipid peroxidation owing to a high polyunsaturated fatty acid content, and low levels of endogenous antioxidants [52]. Consequently, chronic alcohol consumption depletes essential antioxidants in the brain, thereby exacerbating oxidative damage. Studies have shown that alcohol-induced ROS can lead to excitotoxicity, mitochondrial impairment, neuronal dysfunction, and ultimately cell death—all of which contribute to the development of AUD [48,53]. In both the liver and brain, alcohol can also be metabolized through the microsomal ethanol oxidizing system (MEOS), which consists of cytochrome P450 enzymes, primarily CYP2E1, along with NADPH and oxygen, generating significant amounts of ROS in the process [49,54]. Elevated levels of CYP2E1 not only increase ROS production but also trigger endoplasmic reticulum stress and the unfolded protein response. For instance, acute alcohol consumption has been shown to cause significant dissolution of hepatic mitochondrial DNA via CYP2E1-induced oxidative stress—a phenomenon that can be mitigated by inhibitors such as 4 methylpyrazole [55]. Moreover, acute ethanol exposure disrupts mitochondrial integrity, calcium homeostasis, and synaptic vesicle activity in hippocampal neurons [56]. Mitochondria, the “powerhouse” of cellular energy metabolism, serve as both major sources and targets of oxidative stress. Alcohol-induced oxidative stress damages proteins and DNA and impairs mitochondrial function [49,57]. A critical regulator of mitochondrial biogenesis and energy metabolism is peroxisome proliferator-activated receptor gamma coactivator 1 alpha (PGC 1α), which collaborates with transcription factors such as Nrf 1/2 and TFAM to promote the expression of mitochondrial genes and the formation of new mitochondria [58,59]. However, under inflammatory conditions, pro-inflammatory cytokines like TNF α and IL 1β, via NF κB activation, suppress PGC 1α expression, thereby further exacerbating ROS accumulation and oxidative stress [60]. Chronic alcohol exposure not only compromises mitochondrial function but also leads to structural alterations. Animal studies have shown that chronic intermittent alcohol exposure suppresses mitochondrial aerobic respiration by affecting both the ETC and OXPHOS [61]. Alcohol also reduces the expression of mitochondrial fusion proteins while increasing the levels of Dynamin-related protein 1 (Drp1), which promotes excessive mitochondrial fission. Sustained high levels of ROS can further alter mitochondrial permeability by triggering the opening of the mitochondrial permeability transition pore (mPTP) [57]. This event results in the rupture of the outer mitochondrial membrane, the release of cytochrome c, and the activation of caspase-3, ultimately leading to apoptosis [62]. The subsequent loss of mitochondrial membrane potential and ATP depletion creates a vicious cycle of ROS accumulation and further mitochondrial damage [63]. Clinical studies and meta-analyses comparing antioxidant levels in AUD patients with healthy controls have revealed that antioxidant enzyme activities—such as those of SOD and glutathione—are significantly reduced, while markers of lipid peroxidation like malondialdehyde (MDA) are markedly elevated [64,65,66]. These findings underscore the critical role of alcohol-induced oxidative stress in impairing mitochondrial function and contributing to neurodegeneration (Figure 1).

## 4. ROS and AUD

Chronic heavy drinking induces cellular apoptosis and cognitive impairment that lead to white matter atrophy, axonal loss, and demyelination, thereby significantly increasing the risk of developing Alzheimer’s disease, vascular dementia, and other neurodegenerative disorders [48,67,68,69,70,71]. One key mechanism underlying these changes is oxidative stress, which initiates neurodegeneration through a cascade of inflammatory responses. For instance, binge alcohol consumption disrupts the intestinal microbiota and increases the permeability of the intestinal mucosa—a condition commonly referred to as “leaky gut”. This disruption enables pathogen-associated molecular patterns (PAMP), such as bacterial LPS, to translocate into the extraintestinal space and enter systemic circulation [72,73]. Once in circulation, LPS can migrate to organs like the liver and brain, where it activates the TLR4 signaling pathway, an essential component of the innate immune response [7,8,9,10]. Experimental studies have further demonstrated the deleterious effects of LPS on the brain. In one study, intracerebroventricular injection of LPS in mice resulted in a significant decline in neuronal activity, accompanied by cell loss and microglial activation in the hippocampus [74]. Behavioral assessments, including the Morris Water Maze and passive avoidance tests, confirmed that LPS administration impairs cognitive function and memory [75]. Upon binding to TLR4, LPS facilitates the translocation of NF-κB transcription factors into the nucleus, thereby increasing the expression of innate immune genes and allowing for the resultant inflammatory cytokines to cross the blood–brain barrier (BBB) [76,77]. Furthermore, TLR4 activation—either directly by LPS or via cytokine receptor signaling—upregulates the mRNA expression of pyrin domain-containing protein 3 (NLRP3) and pro-IL-1β, setting the stage for the assembly of the NLRP3 inflammasome [78]. Once activated, the NLRP3 inflammasome dissociates from its autoinhibited state, activates caspase-1, and promotes the release of pro-inflammatory cytokines, particularly IL-1β and IL-18, thereby initiating an inflammatory cascade. In addition, there exists a bidirectional feedback mechanism between NLRP3 activation and mitochondrial dysfunction in microglia. Excessive alcohol consumption impairs OXPHOS and alters mitochondrial permeability, leading to increased levels of cytosolic mitochondrial DNA. Owing to its prokaryotic origin, mitochondrial DNA acts as a PAMP that further stimulates inflammasome assembly, while the released cytokines exacerbate mitochondrial damage [48]. Although cytokines and toxic metabolites of alcohol are primarily generated in peripheral organs, such as the liver, heart, lungs, and reproductive organs, they can readily cross the BBB—especially after alcohol consumption—thereby triggering neuroinflammatory and neurodegenerative processes in the brain [10,79]. Supporting this, several studies have demonstrated microglial activation in various brain regions of human AUD patients, including the putamen, external globus pallidus, and ventral pallidum, as evidenced by the upregulation of the microglial activation marker Iba-1. Further research on microglial activation in these regions will deepen our understanding of the pathological mechanisms underlying AUD, not only in the context of alcohol-induced oxidative stress but also concerning alcohol-seeking behavior, relapse, and compulsion (Figure 1) [80].

## 5. ROS and AD

In AD, certain brain regions are particularly vulnerable to damage induced by ROS due to their high metabolic activity. Neurons in the cornu ammonis 1 (CA1) region of the hippocampus, the entorhinal cortex, the temporal lobe, the frontal cortex, and the amygdala require substantial energy and are, therefore, more susceptible to oxidative stress. For instance, the CA1 region not only plays a crucial role in processing sensory and motor signals but also in storing and integrating spatial, contextual, and emotional information before relaying it to other brain regions [81]. As a result, excessive ROS in this region can lead to neurodegeneration, impairing learning and memory [82]. The entorhinal cortex, known as one of the earliest sites of degeneration in AD, exhibits pathological changes, such as nerve fiber tangling and cell death. Given its extensive connections with the hippocampus, damage in the entorhinal cortex can disrupt cognitive processes and memory formation, thereby contributing to the progression of AD [83,84]. Similarly, atrophy in the temporal lobe is among the first structural changes observed in AD and is associated with deficits in memory, language, and vision along with early signs of nerve fiber tangling [85]. The frontal cortex, essential for higher cognitive functions, is also notably vulnerable; its thinning is a characteristic marker of AD-related neurodegeneration [86]. Moreover, the amygdala, which plays a key role in emotional regulation, often shows progressive atrophy in AD. Although initial volume loss may be subtle, ongoing degeneration is associated with olfactory deficits, emotional dysfunction, and neuropsychiatric symptoms, such as depression and hypersensitivity [87,88]. Additionally, the NADPH oxidase (NOX) family contributes significantly to ROS production in various tissues. In particular, NOX2-mediated post-synaptic peroxide production in the CA1 synapses has been implicated in the long-term inhibition of synaptic transmission, further linking oxidative stress to impaired neuronal communication [89,90]. It is also noteworthy that alcohol consumption exacerbates hippocampal damage, leading to rapid aging and atrophy of the CA1 region. This alcohol-induced neurodegeneration contributes to spatial learning impairments and altered emotional behaviors, which may eventually progress toward an AD phenotype (Figure 1) [91,92,93,94].

## 6. Mechanisms of ROS Dysregulation in AD

Oxidative stress is defined as an imbalance between oxidation promoters and antioxidants, which leads to the disruption of the redox circuit and subsequent macromolecular damage [95]. In normal cells, a delicate balance between ROS production and antioxidant defenses is essential, but the brain—requiring large amounts of oxygen and glucose—is particularly susceptible to oxidative damage [96]. In AD, oxidative stress affects nucleic acids, proteins, and lipids within the central nervous system, and as the activity of antioxidant enzymes fluctuates, oxidative damage continues to accumulate. When this balance is disrupted, increased ROS production overwhelms the antioxidant defenses, leading to mitochondrial damage that can occur even in the early stages of AD, prior to the overt pathology of Aβ deposition [97]. Excessive ROS not only induce chronic neuroinflammation but also contribute to mitochondrial dysfunction, nerve cell loss, and protein misfolding [98,99]. Furthermore, heightened ROS levels trigger the toxic processing of amyloid beta precursor protein (APP), which in turn increases the production of Aβ peptides, a hallmark of AD pathology [6,100]. Amyloid-beta-binding alcohol dehydrogenase (ABAD), a mitochondrial enzyme structurally similar to the ADH family, plays a neuroprotective role under physiological stress conditions. However, when ABAD binds to Aβ peptides, its active tetramer form converts into an inactive dimer, leading to the accumulation of toxic aldehydes and ROS. This interaction contributes to mitochondrial dysfunction and promotes further Aβ deposition in Alzheimer’s disease. However, when the ADH binds to Aβ, its active tetramer form converts into an inactive dimer, leading to the accumulation of toxic aldehydes and ROS. This interaction disrupts OXPHOS, further increasing the spontaneous production of H_2_O_2_ and ROS, thereby promoting the aggregation of Aβ in AD patients [5,101]. Given that ABAD is named for its structural similarities to ADH family members, a potential relationship between alcohol consumption and ADH function is plausible, although it has not yet been fully explored. ADH genes, including ADH3 and ADH4, exhibit significant genetic polymorphisms across populations. These variants can influence the rate of ethanol metabolism and the accumulation of acetaldehyde, thereby potentially modulating susceptibility to alcohol-related brain damage and neurodegenerative processes [102]. Recent studies have demonstrated that Aβ interacts not only with ABAD but also with ADH3 and ADH4 [5], suggesting that the involvement of ADH and related ADH family members in alcohol-induced AD pathology remains an important subject for future investigation (Figure 1).

## 7. ROS Play Distinct Roles in AUD and AD

Previous studies have suggested that abnormal alcohol consumption in AUD may negatively impact neurodegenerative processes, increasing the risk for conditions such as AD and dementia. For example, a large cohort study in Sweden reported that approximately 10 million discharged AUD patients were readmitted with alcoholic dementia, implying that alcohol use contributes to neurodegeneration via oxidative stress and other mechanisms [11,12]. Although chronic, high levels of alcohol consumption are well known to promote AD development, recent findings also indicate that low-to-moderate alcohol intake might confer neuroprotective effects during aging, underscoring the need to consider consumption patterns when evaluating its impact [13]. From the perspective of oxidative stress, ROS serve as key drivers of neurodegeneration in both AUD and AD, yet the underlying mechanisms differ between the two. In AUD, chronic alcohol exposure induces oxidative stress primarily through CYP2E1-mediated ethanol metabolism [49], leading to excessive ROS accumulation in neurons and glial cells [1]. This accumulation initiates mitochondrial dysfunction, disrupts mitophagy, and exacerbates excitotoxicity by altering glutamate homeostasis. Furthermore, ROS-induced dysregulation of tight junction proteins in the gut epithelium increases intestinal permeability, allowing for endotoxins such as LPS to enter the systemic circulation. The resulting gut-derived inflammation further amplifies central nervous system oxidative stress via the gut–brain axis, thereby compromising neuronal function [103]. In contrast, AD pathology is largely characterized by Aβ accumulation and tau hyperphosphorylation. The oxidative environment in AD promotes lipid and protein oxidation as well as DNA damage, which accelerates neuronal death [5,104]. Moreover, mitochondrial ROS contribute to the activation of the NLRP3 inflammasome and subsequent neuroimmune responses that lead to neurodegeneration. Emerging evidence suggests that alterations in the gut microbiome and increased permeability also exacerbate oxidative stress and promote systemic inflammation in AD models, paralleling mechanisms observed in AUD [105,106,107]. Another shared mechanism involves glutamate-induced excitotoxicity. Increased extracellular glutamate levels and impaired reuptake have been implicated in tissue damage across various neurodegenerative diseases, including AD, Parkinson’s disease, amyotrophic lateral sclerosis, and Huntington’s disease [108]. In this context, glutamate transporters such as EAAT1 and EAAT2—predominantly expressed in microglia and astrocytes—play a critical role in maintaining glutamate homeostasis. In AUD, post-mortem human brain samples have revealed increased EAAT1 expression, possibly representing a compensatory response to elevated extracellular glutamate, while studies in alcohol-fed rats have shown decreased EAAT2 expression, which may exacerbate excitotoxicity and oxidative stress [109]. Conversely, in AD, both EAAT1 and EAAT2 levels are reduced, potentially contributing to increased Aβ accumulation [110,111]. In addition, chronic alcohol intake has been shown to suppress the expression of glutamate transporter-1 (GLT-1) in various reward-associated brain regions, leading to abnormal extracellular glutamate levels that further contribute to excitotoxicity [108,112,113]. These observations suggest a novel intersection between AUD and AD, where mitochondrial dysfunction, excitotoxicity, and gut–brain interactions converge to drive oxidative stress-related neurodegeneration. In AUD, ethanol directly damages mitochondrial DNA, creating a vicious cycle of increased oxidative stress and diminished cellular defenses [57]. While aging remains the primary risk factor for AD [114], non-aging factors such as elevated Aβ levels can also induce mitochondrial dysfunction and increased ROS production. This leads to abnormal energy metabolism, synaptic dysfunction, and tau phosphorylation, further exacerbating neurodegeneration and ultimately resulting in neuronal death [115]. Understanding these shared and distinct mechanisms of ROS dysregulation in AUD versus AD may provide valuable insights for developing targeted interventions aimed at mitigating the complex pathophysiology of oxidative stress-related neurodegeneration (Figure 1).

## 8. Therapeutic Target

Various studies suggest that mitochondrial-targeted strategies aimed at reducing ROS production hold promise for mitigating diseases such as AUD and AD. One approach involves activation of the antioxidant transcription factor Nrf2. Nrf2 is expressed in most tissues and regulates the expression of cell-protective genes [116]. Nrf2 protein comprises several evolutionarily conserved domains (Neh1–Neh6) [117]; notably, its N-terminal Neh2 domain contains the DLG (Asp-Leu-Gly) and ETGE (Glu-Thr-Gly-Glu) motifs that directly interact with the negative regulator Keap1, facilitating degradation of excess Nrf2 and thereby maintaining cellular homeostasis [118]. Under conditions of oxidative stress, specific cysteine residues within Keap1 become modified, altering its conformation and allowing for Nrf2 to escape degradation; in the brains of AD patients, Nrf2 is predominantly localized in the cytoplasm of hippocampal neurons, where it drives the expression of antioxidant genes to reduce ROS accumulation caused by Aβ peptides [119,120]. Pharmacological upregulation of Nrf2 has shown potential as a therapeutic target for AD, contributing to the alleviation of associated symptoms [121]. Mitochondria-targeted antioxidants represent another therapeutic strategy. MitoQ, a derivative of ubiquinol, accumulates in the inner mitochondrial membrane via its triphenylphosphonium ion moiety and is converted to its active antioxidant form, ubiquinol [122]. MitoQ inhibits lipid peroxidation and the accumulation of ROS primarily produced by Complexes I and III of the electron transport chain; its oxidized form, ubiquinone, is subsequently recycled to its active form by Complex II [123]. In an in vivo study of chronic alcohol abuse, lung tissue from mice treated with MitoQ showed reduced ROS levels compared to those from alcohol-exposed mice without MitoQ, as evidenced by dihydroethidium staining [124]. Moreover, MitoQ suppressed NLRP3 inflammasome activity and IL-1β secretion in these models [124], and similar antioxidant and anti-inflammatory effects were observed in human prostate cells [125]. MitoQ also crosses the BBB, decreases superoxide production (thereby reducing nitrotyrosine, an oxidative stress biomarker of AD) [126], and promotes neuroblastoma cell proliferation by suppressing mitochondrial fission gene expression and enhancing antioxidant enzyme levels [127]. Although some studies have reported potential nephrotoxicity and gastrointestinal side effects—such as increased brachial artery flow-mediated dilation with mild-to-moderate gastrointestinal discomfort during 6-week chronic supplementation and adverse effects (nausea and vomiting) in Parkinson’s patients administered 80 mg MitoQ [128]—a recent study in healthy adults found that acute, high-dose MitoQ (100–160 mg, adjusted by body weight) did not adversely affect kidney functions [129]. It is also worth noting that the effect of MitoQ may vary by biological sex; for instance, a study in rats demonstrated lower superoxide levels in female offspring compared to males after MitoQ administration during fetal hypoxia [130], although other studies, such as those using Sacs gene knockout mice, did not observe gender differences [131]. Another potential antioxidant is SS-31, a synthetic peptide that specifically targets the inner mitochondrial membrane by binding to cardiolipin, thereby stabilizing mitochondrial structure and preventing the formation of the mPTP [132,133]. SS-31 also scavenges ROS through its dimethyl tyrosine residue, forming di tyrosine radicals that reduce LDL oxidation [134]. Additionally, SS-31 has been shown to inhibit NLRP3 inflammasome activity in hippocampal microglial cells, potentially by modulating DRP1, a protein involved in mitochondrial fission and ROS production [135]. Preclinical studies demonstrate that SS-31 is distributed across major organs—including the heart, lungs, liver, skeletal muscles, and kidneys, with the highest concentrations in the kidneys—and is completely cleared through urination [134]. In a phase IIa clinical trial involving patients with atherosclerotic renal artery stenosis, SS-31 administration did not result in significant adverse effects, such as fever, headache, vomiting, hematuria, or allergic reactions, and no changes in serum creatinine or urine cytology were observed within 24 h [136]. Despite favorable safety profiles in early-phase trials, a phase III study in patients with mitochondrial myopathy did not show significant improvements in measures such as walking distance or fatigue scores [137]; however, SS-31 was well tolerated overall. Dosage optimization remains a subject of ongoing research, as one study in patients with heart failure with reduced ejection fraction (HFrEF) reported that 24% of patients receiving a 40 mg injection experienced treatment-emergent adverse events (TEAEs), such as nausea and fatigue, compared to 4.5% at a 4 mg dose [138]. Targeting the NLRP3 inflammasome is another emerging therapeutic strategy, given its activation by ROS and its involvement in both AD and AUD. MCC950 is a potent NLRP3 inflammasome inhibitor that blocks the interaction between NEK7 and NLRP3 by binding to the Walker B motif within the NACHT domain [139,140]. MCC950 not only reduces NLRP3 activity but also lowers ASC protein levels, thereby inhibiting caspase-1 activation and subsequent release of pro-inflammatory cytokines such as IL-1β and IL-18 [141]. In rat models treated with streptozotocin, MCC950 reduced NLRP3 inflammasome complex levels and caspase-1 activation in the dentate gyrus, CA1, and CA3 regions of the hippocampus, with concomitant improvements in spatial learning, episodic memory, and anxiety [142]. However, while several animal studies support the safety of MCC950, some reports indicate potential adverse effects—including elevated expression of the pro-oxidant genes NOX2 and NOX4, increased glomerular nitrotyrosine content, and markers of renal fibrosis [143], as well as conflicting data regarding hepatotoxicity [144,145]—necessitating further investigation. Following MCC950, SB_NI_112 has emerged as another candidate for inhibiting the NLRP3 inflammasome. SB_NI_112 crosses the BBB and acts in multiple brain regions, including the hippocampus, cortex, cerebellum, and brainstem [146]. It functions by inhibiting NF-κB—an upstream activator of NLRP3—and directly suppressing NLRP3 inflammasome activity, thereby reducing pro-inflammatory cytokines (IL-2, IL-1α, and IL-1β) implicated in neuroinflammation [146]. Notably, SB_NI_112 appears effective even in older animals, in contrast to other inhibitors such as OLT1177 [147], and has demonstrated a favorable safety profile across various administration routes (intravenous, subcutaneous, intraperitoneal, and intranasal) without significant pathological changes in kidney, blood, or electrolyte levels [148]. Nonetheless, comprehensive human safety assessments remain necessary. Caspase inhibitors also offer a potential therapeutic approach. VX-765, for example, acts as a potent inhibitor of caspase-1 by modifying the enzyme’s catalytic cysteine residue. Caspase-1, activated by the NLRP3 inflammasome, cleaves pro IL-1β into its active form, IL-1β, thereby triggering neuroinflammation and further ROS production [149]. In vitro studies have shown that VX-765 reduces levels of NLRP3, caspase-1, and IL-1β while promoting cell survival under H_2_O_2_-induced stress [150]. Additionally, animal studies have demonstrated that VX-765 normalizes IL-1β levels, improves spatial learning and memory in the Barnes maze, and alleviates episodic memory impairments in mutant amyloid protein mice across various age groups [151]. In terms of safety, VX-765 has been evaluated in phase IIa clinical trials for epilepsy in a double-blind, randomized, placebo-controlled design. In these trials, 72.9% of participants in the treatment group experienced TEAEs compared to 83.3% in the placebo group; notably, the treatment group’s TEAEs rate was 6%, while the placebo group’s rate was 0% [152]. However, VX-765 was also associated with elevated liver enzyme levels, highlighting the need for further safety evaluation [152]. As our understanding of ROS in AUD and AD expands, several potential avenues for future investigation have emerged. One promising area involves the gut microbiome. Recent studies suggest that disruptions in the gut microbiota—caused by aging, alcohol consumption, or other factors—lead to reduced levels of anti-inflammatory molecules and compromised intestinal integrity [153]. This disruption facilitates the leakage of LPS into the bloodstream, allowing for its transport to the brain via the gut–brain axis. Upon reaching the brain, LPS increases BBB permeability and activates the NLRP3 inflammasome, potentially accelerating AD progression [153,154]. Furthermore, gut dysbiosis may contribute to increased oxidative stress and alcohol dependence by altering the composition of microbiomes that favor ethanol metabolism, ultimately inducing memory loss and neuropsychiatric behaviors [155]. Future research into the development of prebiotics and probiotics targeting ROS, AUD, and AD may offer novel therapeutic opportunities by modulating the gut microbiome, enhancing gut integrity, and preventing LPS translocation, thereby mitigating neuroinflammation and reducing BBB permeability [156]. In summary, therapeutic strategies aimed at reducing ROS—whether by enhancing Nrf2 activity, employing mitochondria targeted antioxidants such as MitoQ and SS-31, inhibiting the NLRP3 inflammasome via agents including MCC950 or SB_NI_112, or using caspase inhibitors such as VX-765—represent promising approaches to mitigating the neurodegenerative processes associated with both AUD and AD. Continued research into these mechanisms along with the exploration of gut–brain interactions are critical for the development of effective, targeted interventions against oxidative stress-related neurodegeneration (Figure 2).

## 9. Conclusions

ROS play a pivotal role in the pathogenesis of both AUD and AD, acting as central mediators of oxidative stress. Excessive ROS production leads to mitochondrial dysfunction, a decline in endogenous antioxidant defenses, and the dysregulation of neuroinflammatory pathways—such as those mediated by TLR4 and the NLRP3 inflammasome—which further promote neurodegeneration. Although AD and AUD exhibit distinct hallmark features (e.g., Aβ plaques and tau tangles in AD versus alcohol-induced neurotoxicity in AUD), they share common underlying mechanisms driven by the overproduction of ROS. Given the critical role of ROS in both conditions, therapeutic strategies that aim to reduce oxidative stress show considerable promise. Mitochondria-targeted antioxidants, such as Mito-Q and SS-31, have demonstrated the capacity to preserve mitochondrial integrity and lower ROS levels. In addition, agents that inhibit the NLRP3 inflammasome (e.g., MCC950) and caspase-1 (e.g., VX-765) offer further potential by mitigating neuroinflammation. Complementary interventions that target glutamate transport dysfunction and gut–brain axis dysregulation also provide additional therapeutic avenues, particularly in the context of AUD-related neurodegeneration. Future research should prioritize the identification of reliable ROS biomarkers for early diagnosis as well as the development of combination therapies that concurrently address oxidative stress and inflammation. Moreover, a deeper investigation into the interplay of genetic, environmental, and lifestyle factors in ROS-related diseases is essential. The advancement of accessible technologies capable of measuring ROS in vivo and in real time would be of significant therapeutic importance, as such tools would enable the early identification and management of elevated ROS levels, thereby preventing the progression of neurodegenerative lesions (Table 1). In summary, this perspective not only deepens our understanding of the common oxidative mechanisms linking AUD and AD but also highlights promising directions for future clinical trials and targeted interventions against ROS-induced neurodegeneration.

## Figures and Tables

**Figure 1 ijms-26-03272-f001:**
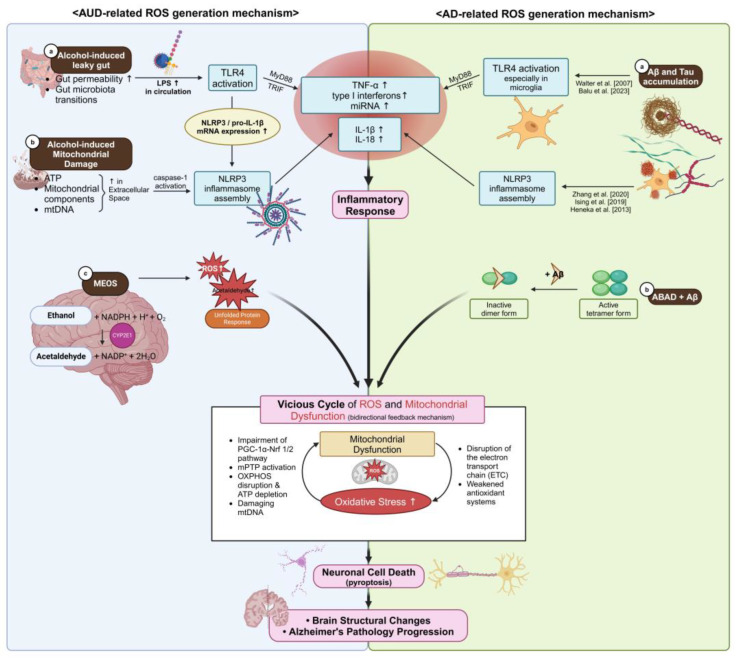
Oxidative stress as a central mechanism linking AUD and AD. The schematic illustrates how oxidative stress and mitochondrial dysfunction contribute to neuroinflammation and neuronal damage in AUD and AD. On the left side, alcohol consumption disrupts the gut barrier, increasing intestinal permeability and allowing for LPS to enter the bloodstream. LPS activates TLR4, triggering NLRP3 inflammasome assembly and promoting the release of pro-inflammatory cytokines and microRNAs(miRNAs). Alcohol-induced mitochondrial damage leads to ATP depletion, mitochondrial DNA (mtDNA) release, and oxidative stress, amplifying neuroinflammation. Chronic alcohol exposure also activates the MEOS in brain tissues, where CYP2E1 catalyzes ethanol metabolism, producing excessive ROS and acetaldehyde, which contributes to unfolded protein response (UPR) and neuronal toxicity. On the right side, Aβ and tau protein accumulation, key hallmarks of AD, activate TLR4 signaling in microglia, leading to NLRP3 inflammasome assembly and inflammatory responses. The interaction between Aβ and ABAD increases ROS accumulation and mitochondrial dysfunction by inactivating ABAD, which usually exists in an active tetramer form. Inflammatory responses in both AUD and AD create a vicious cycle of ROS production and mitochondrial dysfunction, reinforcing oxidative damage through a bidirectional feedback mechanism. Mitochondrial impairment leads to ETC disruption, ATP depletion, and mPTP activation, increasing oxidative stress. As the cycle progresses, neuronal cell death through pyroptosis, brain structural changes, and Alzheimer’s pathology progression accelerate. The entorhinal cortex, known as one of the earliest sites of degeneration in AD, exhibits pathological changes, such as nerve fiber tangling and cell death [14,15,16,17,18].

**Figure 2 ijms-26-03272-f002:**
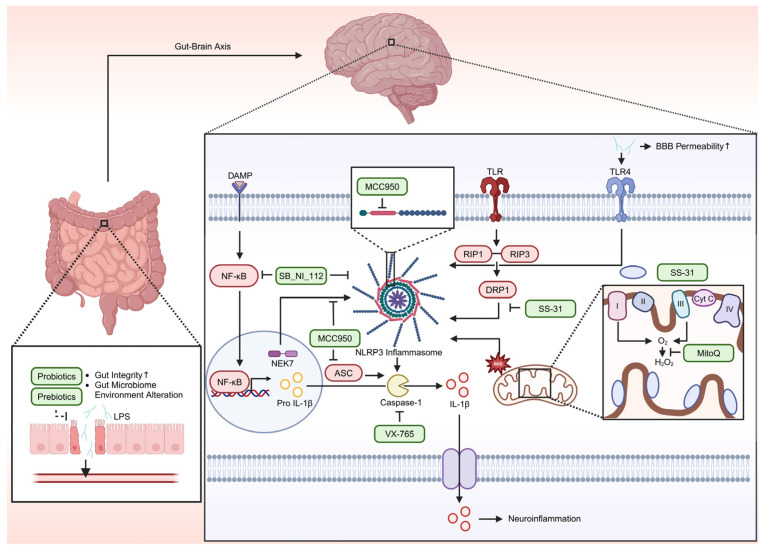
Therapeutic strategies targeting ROS in AUD and AD. The interplay between the gut–brain axis, mitochondrial dysfunction, and neuroinflammation highlights potential therapeutic targets in AD and AUD. In the gut, probiotics and prebiotics help maintain intestinal integrity, regulate the gut microbiota, and prevent LPS translocation. Increased intestinal permeability allows for LPS to enter the bloodstream, activating TLR4 and triggering NLRP3 inflammasome activation, which leads to BBB dysfunction. At the mitochondrial level, MitoQ reduces ROS by targeting the inner mitochondrial membrane, preventing lipid peroxidation and ROS production at Complexes I and III. SS-31 binds to cardiolipin to stabilize mitochondrial dynamics, prevent mPTP formation, and modulate Dynamin-related protein 1 (DRP1). Both compounds inhibit NLRP3 inflammasome activation, thereby mitigating neuroinflammation. Inflammasome inhibitors provide additional therapeutic strategies. MCC950 blocks NEK7/NLRP3 interaction by binding the Walker B motif in the NACHT domain, suppressing caspase-1 activation and downstream inflammation. SB_NI_112 inhibits NF-κB signaling to reduce NLRP3 activation and the production of inflammatory cytokines. VX-765 selectively inhibits caspase-1, preventing IL-1β release and attenuating neuroinflammation.

**Table 1 ijms-26-03272-t001:** Comparison of AUD and AD.

Category	Alcohol Use Disorder (AUD)	Alzheimer’s Disease (AD)	References
Primary Causes	Chronic heavy drinking and alcohol dependence	Aging, heredity (APOE4), environmental factors	[114,157]
Key Pathological Mechanisms	-Increased ROS production via alcohol metabolism -Mitochondrial dysfunction and ATP depletion -Neuroinflammation (TLR4, NLRP3 activation)	-Aβ plaque and tau protein aggregation -Oxidative stress and mitochondrial dysfunction -Neuroinflammation (TLR4, NLRP3 activation)	[2,10,57,59,115]
Neuronal Damage Mechanisms	-Cytotoxic effects from alcohol metabolism -Mitochondrial dysfunction-induced neuronal loss -Activation of inflammasome pathways	-Aβ-mediated cytotoxicity -Mitochondrial dysfunction-induced neuronal loss -Chronic neuroinflammation and synaptic loss	[7,8,12,14]
Affected Brain Regions	Frontal cortex, Hippocampus, Basal ganglia	Frontal cortex, Hippocampus, Temporal lobe, Amygdala	[1,81,87]
Cognitive Impairments	Memory loss, impaired impulse control, reduced attention span	Memory loss, spatial disorientation, impaired judgment	[85,86]
Psychiatric Effects	Depression, anxiety, increased impulsivity	Depression, social withdrawal, personality changes	[88,158]
Therapeutic Strategies	-Antioxidants (MitoQ, SS-31) -Inflammation modulators (MCC950, SB_NI_112) -Neuroprotective treatments under investigation	-Antioxidants (MitoQ, SS-31) -Research on Aβ clearance therapies	[132,142,149]

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
