# Peer review of "Reactive Oxygen Species as a Common Pathological Link Between Alcohol Use Disorder and Alzheimer’s Disease with Therapeutic Implications"

_ijms, 2025, doi:10.3390/ijms26073272_

Round 1

Reviewer 1 Report

Comments and Suggestions for Authors

The authors have addressed an interesting topic in their paper, as finding a common denominator for two neurodegenerative diseases induced and enhanced by ROS production is remarkable. However, the work is very difficult to read. There are multiple repetitions of the same information in a slightly altered way, which creates a feeling of chaos - it is very difficult to follow the main idea of the authors. The work needs to be tidied up and key information completed in a number of places. Given the title of the paper, this paper lacks information on how neurodegeneration (in addition to the inflammation-generating factors mentioned in Figure 1) can be promoted by reactive oxygen species and oxidative cell damage. The reviewed proposed work is conceptually similar to other already published comparative work (e.g. DOI 10.1186/s12929-017-0379-z) - where the mechanism of ROS and oxidative stress in Alzheimer's disease is already described. As far as the metabolism of ethanol is concerned, it is worth supplementing the paper with reliable information on its metabolism. There are three pathways for the oxidation of ethanol (in the liver and brain) by: alcohol dehydrogenase, the microsomal ethanol oxidation system (MEOS), of which cyp2E1 is a part, and catalase. With around 90% of acetaldehyde being produced and metabolized to acetate in the liver by both ADH and MEOS, however, the contribution of MEOS to ethanol oxidation is individually variable (generally small, around 10-20% of the ethanol taken in). Chronic ethanol consumption increases CYP2E1 levels in the liver by approximately 5-10x. The toxic effects of acetaldehyde have been neglected in this paper, and it can, for example, react with catecholamines and with serotonin to disrupt neurotransmission processes, it can compete with other aldehydes in enzymatic reactions and bind to enzymes, including those protecting cells from free radicals, acetaldehyde will also react directly with glutathione, reducing its ability to defend itself by H2O2 and preventing lipid peroxidation. Ethanol itself can interfere with the action potential of nerve cells and thus disrupt the release of neurotransmitters. Nevertheless, as the authors noted, a better understanding of the impact of oxidative stress in AUD and AD and how to manage it may offer new targets for therapy.

Some issues - a few specified, but the text needs overall review and improvement:

  • Page 1, line 25 - the pathological mechanism of both diseases is complex, I recommend replacing the key word with common
  • Page 1, line 39 - add a reference to figure 1
  • Page 1, line 40 – it is: ‘Epidemiological data further support the link between AUD and an 39 increased risk of dementia’ - add a reference
  • Page 2, line 57 - please write the hydroxyl radical in brackets correctly
  • Page 2, line 70-72 – 1) A sentence about alcohol metabolism thrown into the middle of a chaotic description of the role of the mitochondrial electron transport chain in generating ROS; 2) the sentence on alcohol metabolism appears for the first time and does not contain reliable information
  • Page 3, line 93-95 – I cannot agree with the statement that ‘ADH is largely absent in the central nervous system’, please read: Waddell J, McKenna MC, Kristian T. Brain ethanol metabolism and mitochondria. Curr Top Biochem Res. 2022;23:1-13. PMID: 36873619; PMCID: PMC9980429.
  • Page 3, line 99-101 - Under normal physiological conditions, in addition to enzymatic antioxidants (several have been mentioned), there are also non-enzymatic antioxidants (e.g. glutathione) that help to counteract ROS - please include this information.
  • Page 3, line 109-110 – 1) it is: ‘and CYP2E1’ – it should be ‘the microsomal ethanol oxidation system (MEOS)’, which consists of cytochrome P450 (mainly CYP2E1, NADPH and oxygen); 2) this sentence should have appeared earlier in the text
  • Page 5, line 187 – CA1 - please elaborate on the abbreviation
  • Page 5, line 234 – ABAS - please elaborate on the abbreviation
  • Page 6, line 237 – ADH3, ADH4 - early information on the polymorphism of the gene encoding alcohol dehydrogenase is missing
  • Page 7, line 298 – DLG, ETGE motif - please elaborate on the abbreviation – the same situation in line 332, 335 I 383
  • Table 1 - when comparing AUD and AD it would be better to write the same reasons together once for both columns (AUD and AD) instead of creating a separate commonalities line and repeating information already contained

It is worth noting further that of the 154 articles cited in the references section, 86 items are from the last five years and no excessive self-citation was found.

Nevertheless, at this stage I recommend that the publication be rejected. However, the authors are encouraged to resubmit a revised manuscript.

Author Response

Comments 1: Page 1, line 39 - add a reference to figure 1

-Response 1:  Thank you for your valuable comment. We have revised the sentence to include a direct reference to Figure 1, which illustrates the shared pathological mechanisms between AUD and AD, including ROS production, mitochondrial dysfunction, and neuroinflammation.

Comments 2: Page 1, line 40 – it is: ‘Epidemiological data further support the link between AUD and an 39 increased risk of dementia’ - add a reference

-Response 2:  We appreciate the reviewer’s insightful comment. The sentence “Epidemiological data further support the link between AUD and an increased risk of dementia” is supported by references [11,12], which are cited in the following sentence. Both references provide robust epidemiological evidence that individuals with chronic alcohol use disorder have a significantly higher risk of cognitive impairment and Alzheimer’s disease. Therefore, to clarify this connection, we have explicitly associated references [11,12] with this sentence.

Comments 3: Page 2, line 57 - please write the hydroxyl radical in brackets correctly

-Response 3:  Thank you for pointing this out. We have corrected the formatting of the hydroxyl radical to the proper notation “hydroxyl radical (•OH)” in the revised manuscript to ensure scientific accuracy and clarity.

Comments 4: Page 2, line 70-72 – 1) A sentence about alcohol metabolism thrown into the middle of a chaotic description of the role of the mitochondrial electron transport chain in generating ROS; 2) the sentence on alcohol metabolism appears for the first time and does not contain reliable information

-Response 4:  We appreciate the reviewer’s thoughtful comment. We have revised this section to improve logical flow and clarity. Specifically, we have separated the description of mitochondrial ROS production from the introduction of alcohol metabolism. Additionally, we have updated the sentence on alcohol metabolism to include more accurate and evidence-based information, supported by appropriate citations.

Comments 5: Page 3, line 93-95 – I cannot agree with the statement that ‘ADH is largely absent in the central nervous system’, please read: Waddell J, McKenna MC, Kristian T. Brain ethanol metabolism and mitochondria. Curr Top Biochem Res. 2022;23:1-13. PMID: 36873619; PMCID: PMC9980429.

-Response 5:  Thank you for your insightful comment and relevant reference. We agree with the reviewer that the original statement was too strong and did not accurately reflect the presence of alcohol dehydrogenase (ADH) in the central nervous system (CNS). Based on the work by Waddell et al. (2022), which demonstrates that multiple ethanol-metabolizing enzymes, including ADH, are indeed expressed in various brain regions, we have revised the sentence to reflect this more accurately.

Comments 6: Page 3, line 99-101 - Under normal physiological conditions, in addition to enzymatic antioxidants (several have been mentioned), there are also non-enzymatic antioxidants (e.g. glutathione) that help to counteract ROS - please include this information.

-Response 6:  We thank the reviewer for this helpful suggestion. We have revised the sentence to include non-enzymatic antioxidants such as glutathione.

Comments 7: Page 3, line 109-110 – 1) it is: ‘and CYP2E1’ – it should be ‘the microsomal ethanol oxidation system (MEOS)’, which consists of cytochrome P450 (mainly CYP2E1, NADPH and oxygen); 2) this sentence should have appeared earlier in the text

-Response 7:  We appreciate the reviewer’s valuable feedback. In response, we have revised the terminology to correctly refer to the “microsomal ethanol oxidizing system (MEOS),” which includes cytochrome P450 (primarily CYP2E1), NADPH, and oxygen, as suggested. Additionally, to improve the logical flow, we have moved this sentence earlier in the paragraph where ethanol metabolism in the brain is first discussed.

Comments 8: Page 5, line 187 – CA1 - please elaborate on the abbreviation

-Response 8:  Thank you for your comment. We have revised the manuscript to elaborate on the abbreviation “CA1” upon its first mention by writing out “Cornu Ammonis 1 (CA1)”.

Comments 9: Page 5, line 234 – ABAS - please elaborate on the abbreviation

-Response 9:  Thank you for pointing this out. We have revised the manuscript to elaborate on the abbreviation “ABAD” upon its first mention by writing out “Amyloid-beta-binding alcohol dehydrogenase (ABAD)”.

Comments 10: Page 6, line 237 – ADH3, ADH4 - early information on the polymorphism of the gene encoding alcohol dehydrogenase is missing

-Response 10:  We appreciate the reviewer’s insightful suggestion. In response, we have revised the manuscript to include brief information on the genetic polymorphisms of alcohol dehydrogenase (ADH) genes, particularly ADH3 and ADH4, which may influence the efficiency of ethanol metabolism and potentially modulate susceptibility to alcohol-related neurodegeneration. This additional context helps to frame better the discussion on the interaction between Aβ and ADH isoforms.

Comments 11: Page 7, line 298 – DLG, ETGE motif - please elaborate on the abbreviation – the same situation in line 332, 335 I 383

-Response 11:  We thank the reviewer for pointing this out. In response, we have elaborated on the abbreviations “DLG” and “ETGE” motifs upon their first mention to ensure clarity for readers unfamiliar with these terms.

Comments 12: Table 1 - when comparing AUD and AD it would be better to write the same reasons together once for both columns (AUD and AD) instead of creating a separate commonalities line and repeating information already contained

-Response 12:  We thank the reviewer for the helpful suggestion. In response, we have revised Table 1 to eliminate the separate “Commonalities” row. Instead, we directly integrated overlapping features—such as increased ROS production, mitochondrial dysfunction, neuroinflammation, and neuronal damage—into the respective columns under AUD and AD.

Reviewer 2 Report

Comments and Suggestions for Authors

Review of the article “ Reactive Oxygen Species as a Common Pathological Link Between Alcohol Use Disorder and Alzheimer’s Disease with Therapeutic Implications“ by  Hyein Song et al.

The manuscript is organized following the journal's instructions for authors. The abstract provides an adequate overview of this comprehensive review article regarding the main topic of the article. (the role of ROS, especially mitochondrial ROS, in AUD and AD) and therapeutic procedures resulting from it. The manuscript is devided in logical sections throught which the authors gradually elaborate the topic and indicate the similarities and differences between the AUD and AD. The section Therapeutic target in detail suggests the therapeutic strategies based on the molecular changes present in AUD and AD. Therefore, the content of the manuscript is easy to follow. The illustrations, which are visually nice designed and have a very clear content that builds on the content of the textual part of the manuscript, also contribute to this. The references are appropriate. In accordance with the aforementioned, I believe that the manuscript will be of interest to the scientific community.

Author Response

Comments 1: The manuscript is organized following the journal's instructions for authors. The abstract provides an adequate overview of this comprehensive review article regarding the main topic of the article. (the role of ROS, especially mitochondrial ROS, in AUD and AD) and therapeutic procedures resulting from it. The manuscript is devided in logical sections throught which the authors gradually elaborate the topic and indicate the similarities and differences between the AUD and AD. The section Therapeutic target in detail suggests the therapeutic strategies based on the molecular changes present in AUD and AD. Therefore, the content of the manuscript is easy to follow. The illustrations, which are visually nice designed and have a very clear content that builds on the content of the textual part of the manuscript, also contribute to this. The references are appropriate. In accordance with the aforementioned, I believe that the manuscript will be of interest to the scientific community.

-Response 1:  We sincerely thank the reviewer for the thoughtful and encouraging evaluation of our manuscript, “Reactive Oxygen Species as a Common Pathological Link Between Alcohol Use Disorder and Alzheimer’s Disease with Therapeutic Implications.”

We are grateful for your recognition of the manuscript’s structure, clarity, and the visual quality of the illustrations. We especially appreciate your positive comments regarding the relevance of our discussion on mitochondrial ROS, the therapeutic implications, and the balance we have sought to maintain in comparing AUD and AD.

Your supportive feedback reinforces the value of our work and encourages us to continue exploring the pathological links between these disorders. We are confident that the revisions and refinements made in response to the reviewers’ suggestions have further improved the clarity and impact of the manuscript.

Thank you again for your kind and constructive review.

Reviewer 3 Report

Comments and Suggestions for Authors

The authors have revised the article appropriately and need no further revision.

Author Response

Comments 1: The authors have revised the article appropriately and need no further revision.

-Response 1: We sincerely thank the reviewer for the encouraging feedback and are grateful for their thoughtful review of our work.

Round 2

Reviewer 1 Report

Comments and Suggestions for Authors

The authors responded accordingly. The text of the manuscript was revised in a way that greatly improved its readability and clarity. I recommend for publication in the current version.

Some minor editorial points for improvement are listed below:

  • Page 1, line 39 - here is a reference to figure 1, which appears much later in the manuscript – Page 10, line 451 - it might be worth moving this image closer to the first quotation, especially as many times the text, in earlier paragraphs, refers to this image
  • Page 2, line 42 and 58 - please remove the extra space between the reference and the full stop at the end of the sentence
  • Page 2, line 84 and Page 6, line 242 – please, use one form of hydrogen peroxide or H2O2, as both have already been introduced in the text.
  • Page 3, line 97 - please use either alcohol dehydrogenase or ADH - this abbreviation was introduced earlier in the text
  • Page 4, line 160 - a space is missing between the word patterns and the abbreviation
  • Page 4, line 181 - maybe instead of the phrase ‘pathogen-associated molecular pattern’ use the already introduced abbreviation ‘PAMP’
  • Page 6, line 228 - remove bracket closure - no bracket opening in earlier text
  • Page 5-6, lines: 233. 236 237 – there are the following phrases: Aβ deposition, Aβ peptides Aβ-binding ADH and the parts of whole name ‘amyloid-beta-binding alcohol dehydrogenase ‘ explaining the abbreviations are introduced only later in the line – 243 - this can be sorted out a bit
  • Page 6, line 284 - please write in brackets or in the list of abbreviations the full name of the EAAT1 and 2 transporters
  • Page 7, line 307 - please remove the extra space between the words ‘at’ and ‘reducing’
  • Page 7, line 324 - please remove the extra space between the words ‘ROS’ and ‘primarily’
  • Page 7, line 351 - please remove the extra space between the words ‘across’ and ‘major’
  • Page 7, line 354 – there is ‘SS- administration’, whether number 31 has been omitted?

Author Response

Comments 1: Page 1, line 39 - here is a reference to figure 1, which appears much later in the manuscript – Page 10, line 451 - it might be worth moving this image closer to the first quotation, especially as many times the text, in earlier paragraphs, refers to this image

-Response 1: We appreciate the reviewer’s suggestion. To improve the logical flow and enhance readability, we have moved Figure 1 to appear directly after its first reference in the Introduction section.

Comments 2: Page 2, line 42 and 58 - please remove the extra space between the reference and the full stop at the end of the sentence.

-Response 2: Thank you for pointing this out. The extra spaces between the references and the sentence-ending periods have been removed on lines 42 and 58.

Comments 3: Page 2, line 84 and Page 6, line 242 – please, use one form of hydrogen peroxide or H2O2, as both have already been introduced in the text.

-Response 3: We have standardized the terminology throughout the manuscript to use “hydrogen peroxide (Hâ‚‚Oâ‚‚)” at its first mention, and subsequently referred to it as “Hâ‚‚Oâ‚‚” to maintain consistency.

Comments 4: Page 3, line 97 - please use either alcohol dehydrogenase or ADH - this abbreviation was introduced earlier in the text.

-Response 4: We have updated the sentence on line 97 to use “alcohol dehydrogenase (ADH)” at first mention and maintained “ADH” consistently in subsequent uses.

Comments 5: Page 4, line 160 - a space is missing between the word patterns and the abbreviation.

-Response 5: This has been corrected by inserting a space between “patterns” and the following abbreviation on line 160.

Comments 6: Page 4, line 181 - maybe instead of the phrase ‘pathogen-associated molecular pattern’ use the already introduced abbreviation ‘PAMP’.

-Response 6: We have replaced the full phrase “pathogen-associated molecular pattern” with the previously introduced abbreviation “PAMP” for consistency and conciseness.

Comments 7: Page 6, line 228 - remove bracket closure - no bracket opening in earlier text.

-Response 7: The unnecessary closing bracket has been removed on line 228 as suggested.

Comments 8: Page 5-6, lines: 233. 236 237 – there are the following phrases: Aβ deposition, Aβ peptides Aβ-binding ADH and the parts of whole name ‘amyloid-beta-binding alcohol dehydrogenase ‘ explaining the abbreviations are introduced only later in the line – 243 - this can be sorted out a bit.

-Response 8: We revised this section to introduce the full name “amyloid-beta-binding alcohol dehydrogenase (ABAD)” clearly before referring to it as “ABAD” and ensured a smooth introduction of related terms, avoiding abrupt switches between Aβ and ADH-related terminology.

Comments 9: Page 6, line 284 - please write in brackets or in the list of abbreviations the full name of the EAAT1 and 2 transporters.

-Response 9: We have added the full names “Excitatory Amino Acid Transporter 1 (EAAT1)” and “Excitatory Amino Acid Transporter 2 (EAAT2)” in brackets at their first mention and included them in the abbreviation list at the end of the manuscript.

Comments 10: Page 7, line 307 - please remove the extra space between the words ‘at’ and ‘reducing’.

-Response 10: The extra space between “at” and “reducing” on line 307 has been removed.

Comments 11: Page 7, line 324 - please remove the extra space between the words ‘ROS’ and ‘primarily’.

-Response 11: We have removed the extra space to correct the formatting.

Comments 12: Page 7, line 351 - please remove the extra space between the words ‘across’ and ‘major’.

-Response 12: The extra space has been deleted for consistent formatting.

Comments 13: Page 7, line 354 – there is ‘SS- administration’, whether number 31 has been omitted?

-Response 13: Thank you for catching this. We have corrected the phrase to “SS-31 administration” to match the rest of the manuscript and ensure clarity.
